# Diversity of Self-Assembled RNA Complexes: From Nanoarchitecture to Nanomachines

**DOI:** 10.3390/molecules29010010

**Published:** 2023-12-19

**Authors:** Maria A. Kanarskaya, Dmitrii V. Pyshnyi, Alexander A. Lomzov

**Affiliations:** Institute of Chemical Biology and Fundamental Medicine SB RAS, Novosibirsk 630090, Russia; makanarskaya@gmail.com (M.A.K.); pyshnyi@niboch.nsc.ru (D.V.P.)

**Keywords:** RNA, self-limited complex, circular RNA, concatemer, rational design, self-assembly, molecular dynamics of nucleic acids, supramolecular complexes of nucleic acids

## Abstract

New tool development for various nucleic acid applications is an essential task in RNA nanotechnology. Here, we determined the ability of self-limited complex formation by a pair of oligoribonucleotides carrying two pairwise complementary blocks connected by a linker of different lengths in each chain. The complexes were analyzed using UV melting, gel shift assay analysis, and molecular dynamics (MD) simulations. We have demonstrated the spontaneous formation of various self-limited and concatemer complexes. The linear concatemer complex is formed by a pair of oligomers without a linker in at least one of them. Longer linkers resulted in the formation of circular complexes. The self-limited complexes formation was confirmed using the toehold strand displacement. The MD simulations indicate the reliability of the complexes’ structure and demonstrate their dynamics, which increase with the rise of complex size. The linearization of 2D circular complexes into 1D structures and a reverse cyclization process were demonstrated using a toehold-mediated approach. The approach proposed here for the construction and directed modification of the molecularity and shape of complexes will be a valuable tool in RNA nanotechnology, especially for the rational design of therapeutic nucleic acids with high target specificity and the programmable response of the immune system of organisms.

## 1. Introduction

Ribonucleic acids (RNA) nanotechnology is an actively developing area of basic research and applied science over the past twenty years [1,2,3]. It is based on the bottom-up self-assembly of RNA chains or structures at the nanometer scale ensembles [1]. Several approaches for building various 1D, 2D, and 3D structures have been developed [1,3]. The most popular methods for the construction of supramolecular complexes are combining well-characterized or newly developing structural elements such as duplex building blocks, overhanging complementary single strands [4] three-way Junction (3Wj) [1,5,6], kissing loop [7], paranemic crossover [8], RNA origami approach [9,10], and others.

Despite the variety of known structural motifs, the development of new approaches to the rational design and rapid and reliable construction of nanostructures of a given geometry remains an urgent task of modern RNA nanotechnology. It is due to the growing interest in RNA-based constructs in biomedicine. The size, shape, and structure of RNA nanoparticles are important factors in gene regulation, targeting, and modulation of the immune response [11,12]. Moreover, the development of RNA nanomachines using structure-variable RNA has become a useful tool in various synthetic biology applications [9,13,14].

We recently demonstrated the ability to construct a variety of linear and cyclic (self-limited) DNA complexes using a pair of oligonucleotides (Figure 1a,b) [15]. This approach is based on the self-association of a pair of oligonucleotides with two transposed complementary blocks. These duplex-forming blocks can be linked through a nucleotide or non-nucleotide linker or do not have a linker. Assembling to the self-limited complexes of defined size and molecularity, concatemers, or their mixtures can be regulated by variation of the length of duplex and loop-forming blocks. These give new opportunities for DNA nanotechnology, research, and nucleic acids therapy.

The proposed approach for the self-assembly of linear concatemer 1D and self-limited 2D complexes can be transferred to RNA nanostructures. This simple construction method, in contrast to sophisticated approaches currently used (e.g., [10,16,17]) will be a useful tool to speed up the research time of new RNA tools by the de novo nanoconstructions design approaches. At the same time, RNA nanotechnology differs significantly from DNA nanotechnology [1]. This difference originated from the fundamental physicochemical properties of these nucleic acids: base pairing, base arrangement, thermodynamic stability, duplex size and geometry, flexibility, variety of secondary structures, etc. [18,19,20,21,22]. Thereby, the approaches developed for DNA cannot be transferred directly to RNA nanoconstructions.

In this work, we analyzed the possibility of self-limited complex formation by a pair of oligonucleotides, determined their molecularity, and showed the diversity of the structures. For the future implementation of RNA self-limited complexes into biomedical and synthetic biology applications, we demonstrated a method for reversible converting circular RNA complexes into linear ones and vice versa.

## 2. Results

### 2.1. Oligonucleotides Design

Our previous analysis showed a variety of DNA self-limited and concatemer complexes formed by a pair of oligonucleotides carrying two pairwise complementary regions connected by linkers of different lengths in each strand [15]. Here, we used similar RNA sequences containing two oligonucleotide blocks connected via an oligouridine linker (Table 1). The M- and N-type oligomers contain two of 10 nt long pairwise complementary blocks (Table 1, Figure 1). Oligomers without linkers (M, N), or with oligouridine linker Ui in M-Ui (i = 1, 2, 3, 5, 7, 10, 15) and N-Uj (j = 1, 2, 3) was synthesized and purified by gel electrophoresis. Thus, we analyzed the effect of the linker in the M- and N-type chains on the type and size of the complex formed.

### 2.2. RNA Complexes Thermal Stability Analysis

Previously, we showed the influence of the temperature on the complex type, geometry, and molecularity [15,23,24]. We demonstrated that the determination of the hybridization thermodynamic parameters is hindered. Therefore, for multicomponent complexes, the UV-melting analysis was used to determine the melting temperature (Tm) of the RNA structures. Tm value was determined as the maximum of the first derivative of the melting curve at 260 and 270 mn and averaged. A hysteresis between the heating and cooling curves was observed. This originated by RNA hydrolysis in the presence of Mg^2+^ ions at high temperatures (for example, [25]). We confirmed this suggestion by Reverse Phase High-Performance Liquid Chromatography (RP-HPLC) of the oligonucleotide mixture before and after the UV melting experiment (Appendix A). Therefore, only denaturation curves were used for reliable Tm determination. UV-melting curves have an s-shape transition, which corresponds to the peak on differential melting curves (Appendix A). The Tm values of the studied complexes are presented in Table 2.

The Tm values are in the range from 50 to 60.7 °C. These values are higher than those observed by us for similar DNA complexes having melting temperatures in the range 40–50 °C. It is in line with well-known data thermal stability of NA duplexes: RNA/RNA > DNA/DNA [18,21,26]. The highest Tm values (58–60.7 °C) were observed for the RNA complexes formed by at least one oligomer without linker: M- Ui/N (i = 1, 2, 3, 5, 7, 10, 15), M/N-U1, and M/N-U2. All other complexes have reduced thermal stability by 50–57 °C. The decrease in thermal stability at the Uj linker introduction is associated with the disruption of the stacking interaction at the junction of duplex structures, which stabilizes the complexes (Figure 1a). It is also confirmed by the asymmetric shape of differential melting curves, which is typical for the tandem [27] or the concatemer [24] complexes with cooperative stacking between duplexes.

The Tm values of 56–58 °C have been determined for complexes M-Ui/N-U2, where i = 1, 5–15. It is most probably originated by retaining highly efficient cooperative stacking between the duplex blocks [28,29]. It is in good agreement with the asymmetric shape of the UV-melting curves concerning nucleic complexes without cooperative stacking in the nicks (Appendix A).

The obtained data on the thermal stability of the complexes indicated that at the temperature of 20 °C or lower, the complexes are fully formed. We selected this temperature for determining complexes’ molecularity by gel shift assay analysis.

### 2.3. Gel Shift Assay Analysis

Gel shift assay analysis of the complexes was performed to determine the type and molecularity of complexes formed by oligonucleotides M-Ui (i = 0, 1, 2, 3, 5, 7, 10, and 15) and N-Uj (j = 0, 1, 2, and 3) (Table 1). Experiments were performed in non-denaturizing conditions, and a double-stranded (ds) DNA ladder was used to determine electrophoretic mobility (see Methods section). The use of DNA is driven by the need to compare previously obtained DNA data with the RNA complexes analyzed here. All complexes were prepared at 10 μM concentration by mixing an equimolar quantity of oligomer M- and N-series at 20 °C. A typical electropherogram is shown in Figure 2.

A various set of bands was observed for complexes with different linkers in oligonucleotides. Distributed trace of stained RNA complexes corresponding to the concatemer can be found for the complexes M-Ui/N (i = 0, 1, 2, 3, 5, 7, and 10), M/N-Uj (j = 1, 2, and 3), and M-Ui/N-Uj (i, j = 1, 2, 3) with exception of M-U3/N-U1 (Figure 2 and Appendix A). A similar observation was made for DNA concatemers [15,23].

For a number of these complexes formed by M-Ui with i > 0, well-defined additional bands with mobility in the range of 200–300 bp of dsDNA were found (Figure 2 and Appendix A). These bands correspond to the self-limited complexes of various molecularities (see blow).

All other complexes have single or multiple well-defined bands on the electropherogram (Figure 2 and Appendix A). The data on complex mobility is summarized in Appendix A. The complex mobility can be collected into three series: ~40–100 bp, ~200–260 bp, and 300 bp and over. DNA complexes of similar mobility contain two, four, or more oligonucleotides in the structure, respectively [15]. To determine what type of complexes the bands with a given mobility belong to and to determine their molecularity, we conducted additional studies.

Previously, we have shown that the molecularity of the complexes determined by gel shift assay should be verified [15]. This is due to the fact that the mobility of a self-limiting complex depends not only on the size of the complex but also on a number of other factors. To confirm the molecularity of the complexes previously proposed by us approach was used [15]. It assumes adding an oligonucleotide-opener (opener) (O) to the mixture of M- and N-series oligonucleotides to determine the self-limited (cyclic) complexes’ molecularity (Figure 1c). Interacting the O with the circular complexes resulted in the formation of linear structures by the strand displacement mechanism. The self-limited complexes consisting of 2 × k oligonucleotides are revealed to be linearized complexes with a different number of molecules from 2 × k + 1, …, 2 × k + 1 − 2 × m, …, 3, where m is in the range from 1 to k − 2. For example, only one linear structure will be observed at a bimolecular complex unfolding by adding opener O, while two additional bands can be found on the gel image for circular tetramer. This approach allows us to determine the number of molecules in the self-limited complexes by counting the number of additional bands and analyzing their mobility in gel-shift assay [15].

At the design of the oligomer O, a pentaadenylate block at 5′-end was introduced. It increases the thermodynamic stability of the complex M-Uj/O by interaction with the Uj linker. The high affinity of O to M-Ui and the presence of linker Ui as an overhang allow the linearization of circular complexes. The UV melting analysis confirms the increase of Tm from 51.8 for M10/O to 60.8 °C M-U5/O associated with the interaction of pentaadenylate of O with a pentauridine block of M-U5 (Appendix A, Appendix A).

To confirm the self-limited complex formation and determine the complexes’ molecularity, we selected complexes with mobility of 80 bp (M-U5/N-U3) and 220 bp (M-U5/N-U1), forming presumably bi- and tetramolecular complexes, respectively. Adding the DNA opener (DO) 5′-d(CCATCATATGAAAAA)-3′ in different concentrations ([DO]:[RNA complex] = 0, 0.1, 0.25, 0.5, 0.75, 1, 2, 5, or 10) to the RNA complexes does not change the complexes distribution and mobility on electropherogram up to 10-fold excess of DO (Appendix A). It indicates the absence of the overhangs in these complexes, which can interact with DNA and change the mobility of the complexes. The impossibility of the DNA opener interaction with RNA complexes by strand displacement mechanism (Figure 1c) originated by lower affinity in DNA/RNA duplexes than in the RNA/RNA in self-limited complex [18,21,30]. It was additionally confirmed by UV-melting analysis (Appendix A). The Tm value of the most stable of the two hybrid complexes DO/N-U3 is 44 °C, whereas the Tm is 50 °C for complex M-U5/N-U3.

Adding the RNA opener O (5′-CCAUCAUAUGAAAAA-3′) to the complexes in different concentrations ([O]:[RNA complex] = 0, 0.1, 0.25, 0.5, 0.75, 1, 2, 5, or 10) lead to the changes in the gel image both for M-U5/N-U3 and M-U5/N-U1 complexes (Figure 3, Appendix A).

The thermal stability of the complex of M-U5 and RNA-opener (M-U5/O, Tm = 60.8 °C) is higher than the M-U5/N-U3 (Tm = 50.0 °C). As a result, complex M-U5/N-U3 (mobility ~80 bp DNA) transforms in the presence of O. Increasing O concentration resulted in the presence of only one additional band corresponding to the trimolecular complex M-U5/N-U3/O of lower mobility (~220 bp) at increasing opener concentration (Figure 3 and Appendix A) as schematically illustrated in Figure 1c. At a stoichiometric ratio of oligonucleotides in the mixture of M-U5, N-U3, and O, the band corresponding to the self-limited complex disappears. Further increase in O concentration up to ten-fold excess, the only low mobility band (~220 bp DNA), presents in the electropherogram. This result corresponds to the bimolecular complex formation by M-U5 and N-U3. Moreover, the bands of three molecular M-U5/N-U3/O complexes are less diffuse than the M-U5/N-U3, confirming the formation of a more rigid linear structure at opener adding. We previously observed a similar electropherogram for the bimolecular DNA/DNA complex [15].

After adding opener (O) to M-U5/N-U1 or M-U3/N-U2 complexes’ band redistribution was observed (Appendix A). A M-U5/N-U1 and M-U3/N-U2 complexes have a pronounced band with mobility of ~220 bp DNA ladder. A new band of higher mobility (mobility ~180 bp) corresponded to the three molecular complexes M-U3/N-U2/O and M-U5/N-U1/O formation can be found at a concentration excess of O (Appendix A). Thus, M-U5/N-U3 complex with a mobility of 80 bp is bimolecular. Complexes with similar mobility and longer linkers, M-U7/N-U3, M-U10/N-U3, and M-U15/N-U3 (Figure 2, lanes 6–7), having a mobility of 90–100 bp similarly corresponded to bimolecular complexes. The presence of a lower mobility band with a cut band at 220 bp indicates the absence of overhangs, allowing the formation of concatemers with diffuse filled lanes. This confirms the formation of self-limiting tetramolecular complexes similar to those demonstrated for M-U5 and N-U1. For the M-U2/N-U3 complex, a highly diffuse band with a mobility of ~300 bp was observed. The presence of a band with a sharp boundary at the top indicates the formation of a high-molecular complex with low stability (Figure 2, lane 3, type IV complex).

The mobility and size of all studied by gel-shift assay analysis complexes M-Ui/ N-Uj (i = 0, 1, 2, 3, 5, 7, and 10; j = 1, 2, and 3) are presented in Appendix A. We summarize these results as a heat map table (Figure 4). The Un (n = 0, 1, 2, 3, 5, 7, 10, and 15) linkers introduction in the middle of oligomers M and N strongly affects the type and molecularity of complexes. The absence of the linker in one or both chains of M- and N-type oligonucleotides accompanied by stacking in the nicks of complexes resulted in the formation of a concatemer or mixture of concatemer and self-limited structures of different sizes. Moreover, for the M-U1/N-U1, M-U1/N-U2, and M-Ui/N-U3 (i = 0, 1, 2, and 3) complexes, the formation of the polymer complexes was found. Complexes M-Ui/N-U2 for i = 1, 2, or 3 predominantly forms self-limited tetramolecular complexes (M-Ui/N-U2)_2_. Complexes M-Ui/N-U2 and M-Ui/N-U3 for i = 5, 7, 10, or 15 and M-U15/N-U1 are bimolecular structures. All other complexes are a mixture of self-limited complexes of different molecularities. These results are in good agreement with UV melting data on thermal stability and shapes of differential melting curves. In addition, the mobilities of the exact size self-limited complexes with the same molecularity formed by RNA and DNA (founded by us previously [15]) are in the same ranges. Bimolecular complexes have mobility below 100 bp dsDNA ladder, tertamolecular is in the range 120–300 bp, and complexes of higher molecularity run are slower than 220 bp dsDNA ladder.

The possible types of complexes, depending on the length of the linker, are the same for DNA and RNA. For example, concatameter complexes are formed in the absence of linkers in both chains or with a size of linker equal to 1 nt. If one of the chains has a linker that is two or more nt long, in the case of RNA, a mixture of concatameric and self-terminating complexes is often formed. In this case, DNA usually forms self-limiting complexes with a molecularity that depends on the length of the linker. Samples in which predominantly one type of complex for RNA is formed are M-Ui/N-Uj, with i five and over and j over one. Typically, bimolecular complexes of DNA and RNA are formed if the linker in the M-chain chain has a length of five or more nucleotides and two or more nucleotides in the N-chain. This is enough to ensure that there are no steric hindrances to the formation of a V-shaped structure.

Founded diversity of the RNA complexes formed by a pair of oligonucleotides with two pairwise complementary blocks connected by linkers of different lengths in each strand gives new opportunities for RNA nanostructure design.

### 2.4. MD Simulation and Analysis

To elucidate the molecular origin of the oligonucleotide linker length influence on the type and molecularity of complex formed, we carried out a series of molecular dynamics (MD) simulations. We have analyzed many complexes of different molecularities: dimer, tetramer, and hexamer. The gel shift assay indicated the formation of the bimolecular structure of the M-U5/N-U2 complex, M-U3/N-U2—a tetramolecular structure, M-U3/N-U3—a set of tetra-, and hexamolecular structures. We built molecular models of the self-limited complexes: dimolecular—M-U5/N-U2, tetramolecular—(M-U3/N-U2)_2_ and (M-U3/N-U3)_2_, hexamolecular—(M-U3/N-U3)_3_. The structure and dynamics were analyzed. To establish the molecular origin for the impossibility formation of a bimolecular complex by M-U3/N-U2, this complex was built and studied in the same manner. The molecular models were built in UCSF Chimera. MD simulations of 1 μs or 500 ns for hexamer complex long were performed in an explicit water shell using the Amber20 software package. The obtained trajectories were analyzed via the cpptraj tool. The root means square deviation (RMSD) of duplex blocks along whole trajectories shows the stability of all studied structures (Appendix A). In contrast, the RMSD values of whole complexes are significantly higher. The flexibility of the complexes increases for complexes of higher molecularity (Appendix A). It is in good agreement with the complexes’ bandwidth found in gel shift assay analysis, which is broader for complexes with higher molecularity.

Detailed analysis of the last 100 ns MD trajectories of bimolecular complexes M-U5/N-U2 and M-U3/N-U2 indicate disruption of one and two base pairs in the complexes, respectively. This is due to the short length of the N-U2 linker, which makes it difficult to arrange the two duplexes in parallel in the complex. The destruction of the two terminal base pairs is thermodynamically unfavorable and becomes the driving force promoting the formation of complexes of higher molecularity.

Tetramolecular complexes (M-U3/N-U2)_2_ and (M-U3/N-U3)_2_ are a combination of rigid duplexes connected by flexible single-stranded RNA linkers. Most duplex blocks have paired all nucleobases along the trajectories. Founded terminal base pair disruption (base pair fraying) in some duplex blocks was also determined previously in computer simulations [31,32].

Hexamolecular complex (M-U3/N-U3)_3_ has well-formed six RNA duplexes with terminal base pairs frying along the trajectory. The extremely high flexibility of this complex found by RMSD analysis (Appendix A) is due to the absence of any steric hindrance in the complex.

Hierarchical cluster analysis indicated the presence of several conformations with a high impact on the trajectory. The most represented in the trajectory’s structures are shown in Figure 5.

Dimeric RNA complexes M-U3/N-U2 and M-U5/N-U2 have a v-shaped geometry of duplexes arrangement. It differs from our previous data for DNA duplexes, which showed a predominantly parallel arrangement of duplexes [15]. These findings are due to the larger width of RNA duplexes than DNA [22]. Tetramolecular complexes (M-U3/N-U2)_2_ and (M-U3/N-U3)_2_ complexes have the shape of a distorted quadrilateral in three-dimensional space due to the single-stranded oligouridine linkers. The hexamer (M-U3/N-U3)_6_ has a shape close to a regular hexagon due to the large number of duplex blocks and flexible linkers in the structure. 

Ten superimposed structures were obtained by cluster analysis of MD trajectories for every complex presented in Appendix A. It clearly shows the flexibility of the complexes, which rises with the complex size, which is in line with the RMSD values of whole complexes.

The MD simulation and analysis indicate that the formation of bimolecular complexes with short linkers originated from the steric hindrance of parallel duplexes orientation. It resulted in energetically unfavorable conformation with disrupted base pairs in the duplex blocks. It results in the formation of complexes of higher than two molecularities. The analyzed tetra- and hexamolcelcular complexes have a high flexibility of the structure due to the presence of flexible single-stranded Un linkers.

### 2.5. RNA Nanomaachines. Topology Regulation

The ability to change and control complexes’ shape and molecularity is an important task in nucleic acid nanoarchitectonics and nanomachine building. Here and in our previous works [15,24], we have shown the possibility of purposefully changing self-limited complexes to linear ones or controllable ones by reducing the size of concatemer complexes by adding an opener oligonucleotide. We formulated principles for the efficient regulation of this process.

The possibility of the reverse process of closure into the cyclic structure of a linearized self-limited complex is not clear. We hypothesize that the addition of a closing oligonucleotide (closer, C) that is complementary to the opening oligomer (O) to the open complex will result in the formation of the initial self-limited complex (Figure 1d). The thermodynamically and kinetically efficient interaction of C with O should result in self–limited complex formation.

To verify the hypothesis, self-limited complexes of different molecularity were chosen: bimolecular (M-U5/N-U2, M-U5/N-U3) and tetramolecular (M-U1/N-U2, M-U2/N-U2, M-U3/N-U1, and M-U5/N-U1). The first experiments showed that the efficient interaction of opener (O) with self-limited dimer complex M-U5/N-U3 resulted in the formation of linear trimolecular complex M-U5/N-U3/O (Appendix A). The addition of closer C at the different concentrations up to two-fold excess led to the formation of linear tetramolecular complex M-U5/N-U3/O/C with mobility slightly higher than trimolecular M-U5/N-U3/O. At increasing concentration of C after reaching concentration ratio [O]:[C] = 1:1, a small amount of closed complex can be found. This is originated by a more thermodynamically favorable interaction of O with trimolecular complex than in O/C duplex. In detail, the formation of additional stacking in the nick between N-U3 and O in the M-U5/N-U3/O complex increases the thermodynamic stability of trimolecular complexes similar to the concatemers described before. Therefore, for the closing of the linear trimeric complex, the affinity of O and C should be increased without the rise of stability of M-U5/N-U3 with O.

For this purpose, we elongate both opener and closer (oligonucleotides CL and OL in Table 1). The sequences were chosen to avoid the formation of intramolecular or self-complementary complexes using the RNA Structure Web Server [33] and to ensure high thermal stability of the OL/CL duplexes [34]. The Tm value of the CL/OL complex is 73 °C higher than the M-U/OL (69 °C). It indicates a favorable interaction of an elongated closer with an elongated opener, then with an opened complex.

Gel shift assay analysis shows that bimolecular complexes M-U5/N-U3 efficiently open by adding of equimolar amount of elongated opener OL (Figure 6a, line 2; Appendix A, lane 7). The linear M-U5/N-U3/OL complex (Figure 6a, line 2, complex II) has lower mobility in the gel than self-limited. Adding a similar quantity of CL removes OL from trimolecular complexes returning self-limited M-U5/N-U3 complexes with the mobility in gel similar to CL/OL (Figure 6a, lines 3, 4, complexes I, V). Small-quantity tetramer linearization and reverse cyclization can be found in Figure 6a. The same observations were found for the M-U5/N-U2 bimolecular self-limited complex.

The possibility of tetramolecular complexes M-U5/N-U1, M-U3/N-U1, and M-U2/N-U2 opening and closing were analyzed by adding OL and CL and analyzing the results by gel-shift assays (Figure 6b, Appendix A).

The addition of elongated opener OL at equimolar or two-fold excess quantity to M-U5/N-U1 resulted in the formation of linear complexes M-U5/N-U1/OL (Figure 6b, lanes 2 and 3). Following the addition of 1x quantity of elongated closer CL led to the formation of a set of complexes: self-limited tetramolecular (M-U5/N-U1)_2_, linear pentamolecular (M-U5/N-U1)_2_/CL, trimolecular M-U5/N-U1/OL and duplexes OL/CL (Figure 6b, lanes 4). A maximum quantity of tetramer and dimer can be found in the mixture, containing equal amounts of OL and CL (Figure 6b, lanes 5). Further increases in concentration CL do not change complexes’ distribution significantly (Figure 6b, lanes 6, 7). Similar results were observed for M-U3/N-U1 and M-U2/N-U2 complexes (Appendix A).

These results, obtained for a series of bi- and tetramolecular self-limited complexes, indicate the possibility of reversible changes in their shape and molecularity.

## 3. Materials and Methods

### 3.1. Materials

All solvents and other reagents were purchased from Sigma at the highest available grade and used without purification. Milli-Q water was used for the synthesis and purification procedures. Oligonucleotides were purchased from the Synthetic biology laboratory ICBFM SB RAS, Novosibirsk, Russia.

### 3.2. Oligonucleotide Synthesis

The DNA oligonucleotides were obtained from the High-accuracy DNA/RNA Synthesis Core of ICBFM SB RAS (Novosibirsk, Russia). It was synthesized using the β-cyanoethyl phosphoramidite method and commercially available synthons (Glen Research, Sterling, VA, USA) on an ASM-800 synthesizer (Biosset, Novosibirsk, Russia) and purified by gel electrophoresis.

### 3.3. Oligonucleotide Concentration Determination

Oligonucleotide concentration was determined by the UV spectroscopy method on a UV-2100 UV-vis spectrophotometer (Shimadzu, Kyoto, Japan). The extinction coefficient at 260 nm was calculated as described in [35]. 

### 3.4. Reverse Phase HPLC Analysis

Reverse Phase High-Performance Liquid Chromatography (RP-HPLC, Econova, Novosibirsk, Russia) analysis was performed on a Milichrom A02 chromatograph (Econova, Novosibirsk, Russia) using column (2 × 75 mm) with ProntoSIL-120-5-C18 sorbent (Econova, Russia) and a gradient of acetonitrile content (0–30%) in a 0.02 M aqueous solution of triethylammonium acetate for 30 min (flow rate 100 µL/min, thermostat temperature 25 °C). Detection of chromatographic peaks was carried out at wavelengths of 260 nm. Wavelength 300 nm was chosen as the baseline.

### 3.5. UV Melting Analysis

UV melting analysis was performed on a Cary 300 Bio spectrophotometer (Varian, Belrose, Australia). The wavelengths of 260, 270, and 330 (baseline) nm were used for analysis [36]. The buffer containing 100 mM NaCl, 10 mM C_2_H_6_AsO_2_Na (sodium cacodylate), 15 mM MgCl_2_, pH 7.2 was used for analysis. The concentration of every oligomer in the solution was 1 μM. Heating and cooling experiments were carried out in the temperature range of 5–95 °C with a gradient of 0.5 °C/min using the 0.2 cm quartz cuvettes. 

### 3.6. Gel Electrophoresis

Electrophoretic analysis was performed in non-denaturing conditions in 15% PAAG (acrylamide and *N*,*N*′-methylene bisacrylamide in a ratio of 39:1) in the buffer TB (89 mM tris(hydroxymethyl)aminomethane, 89 mM boric acid, pH 8.3) and 15 mM MgCl_2_ by applying a voltage of 15 V/cm in magnitude. Gels were thermostated at 20 °C by a water bath using a CC3 thermostat (Huber, Berching, Germany). Xylene cyanol was utilized for visual tracking of RNA migration. Stains-all (Sigma, Auburn, WA, USA) was used to stain the gel. A double-stranded DNA ladder 50–1000 bp (SibEnzyme, Novosibirsk, Russia) was used for the analysis of complexes’ mobility. Samples of 10 μL were prepared in a buffer of 10 mM sodium cacodylate, 100 mM NaCl, and 15 mM MgCl_2_. A concentration of M- or N-series oligonucleotides was 10 μM.

### 3.7. MD Simulation and Analysis

Molecular dynamics (MD) simulations of oligonucleotide complexes were performed as described previously [37]. Complexes were constructed using the UCSF Chimera software v. 1.15 [38]. A-form RNA double helixes were built and manually changed to self-limited complexes. AMBER force field OL3 [39], OPC water [40], and Li/Merz ion parameters [41] for ions in OPC water were used. The MD simulations were performed in the explicit water model (OPC model) in a cuboid box (12 Å) via the pmemd.CUDA program of Amber20 [42]. The sodium ions were added to RNA complexes to neutralize the box net charge. A trajectory of 1 μs long was obtained for complexes. For the hexamolecular complex an MD trajectory of 500 ns was obtained. Analysis of trajectories was performed using the cpptraj tool [43] of AmberTools20. A hierarchical cluster analysis was used to determine the most represented structure in the trajectory. Molecular visualization was prepared using UCSF Chimera software v.1.15 [38].

## 4. Conclusions

RNA has a wide variety of secondary and tertiary structures, for example, duplexes, triplexes [44], quadruplexes [45], loops [46], hairpins [46], kissing hairpins [47], pseudoknots [48], kinks [49], paranemic motifs [8], etc. Their formation is determined by the primary structure and the conditions in which they are found. Such structures can become building blocks for creating RNA nanostructures. However, the development of novel tools for constructing supramolecular complexes is an important task of RNA nanotechnology.

RNA folding is different from DNA folding. The double-stranded RNA helix geometry is A-form under near-physiological conditions, whereas DNA is B-form [22]. This originated by sugar puckering and additionally resulted in higher thermostability of RNA than DNA [50,51]. Moreover, the single-stranded RNA have is large repertoire of noncanonical base pairing than the DNA [52,53]. Thus, not all of proposed methods for DNA nanostructures building can be transferred to RNA complexes and vice versa. Consequently, it is necessary to verify the method for the formation of self-limiting complexes proposed in our previous work [15].

Here, we demonstrate the ability of spontaneous self-limited RNA complex formation by a pair of oligoribonucleotides carrying two pairwise complementary regions connected by linkers of different lengths in each strand. The linear concatemer complex formation by a pair of oligonucleotides without a linker, at least in one of them, was discovered. The molecularity of bi- and tetramolecular complexes was confirmed by analyzing the opening of self-limited complexes by toehold strand displacement method. In some cases, we observed the formation of complexes of various types and/or sizes. Complexes of oligomers with similar nucleotide linker lengths formed by RNA and DNA can differ in complex type and molecularity.

The structure and stability of the studied complexes along the MD trajectories indicate the reliability of the complexes’ structure and dynamics simulation. The dimeric complex with short nucleotide linkers, which was not found in experimental studies, indicates base pair disruption originated in steric hindrances of the close location of the duplexes. MD data on the structure and dynamics of RNA complexes are in good agreement with experimental observations and in line with those obtained for DNA by our group [15,24,54]. The last studies showed a good agreement of the structure and dynamics of spin-labeled self-limited DNA complexes performed by EPR analysis and MD simulations.

Our result shows an easy way to construct RNA complexes of various geometry in comparison to another approach, e.g., 3WJ. In our approach, one or more oligoribonucleotides contain small-size structural motifs: duplex and linker block. During the self-association, they form complexes of desired molecularity and geometry. Produced 2D circular complexes can be linearized into 1D complexes. This process is reversible and could be realized in the organism or the cell. 

The proposed approach for shape and size control can be extrapolated for analogs DNA/DNA and DNA/RNA complexes. The difference in the structure of the DNA/DNA and RNA/RNA complexes indicates the necessity of the analysis for hybrid DNA/RNA complexes.

The results obtained can be used in many applications. For example, linear and various molecularity self-limited complexes can be used in RNA nanoarchitectonics or as building blocks. It is originated by the ability to add dangling ends into the complexes of M- and/or N-series oligomers for functionalization. For example, in designing nanoconstructions for biomedicine, it is possible to attach aptamers, riboswitches, siRNAs, ribozymes, G-quadruplexes, etc [1,30]. On the other hand, dimers, tetramers, or hexamers can be used as an element of higher order RNA nanoconstructions connected via dangling ends similar to the three-way junction or kissing loop mechanism, for example, by hand-to-had or foot-to-foot interaction [1]. As a result, complexes of desired molecularity and geometry or dendrimer constructions can be developed.

We believe that the proposed approach for reversible changes in molecularity and shape of complexes will be a valuable tool for the rational design of therapeutic nucleic acids highly targeted with a programmable response of the immune system of organisms [11,12,55]. The results obtained in this paper open up new perspectives and abilities for RNA nanotechnology and RNA nanomachines.

## Figures and Tables

**Figure 1 molecules-29-00010-f001:**
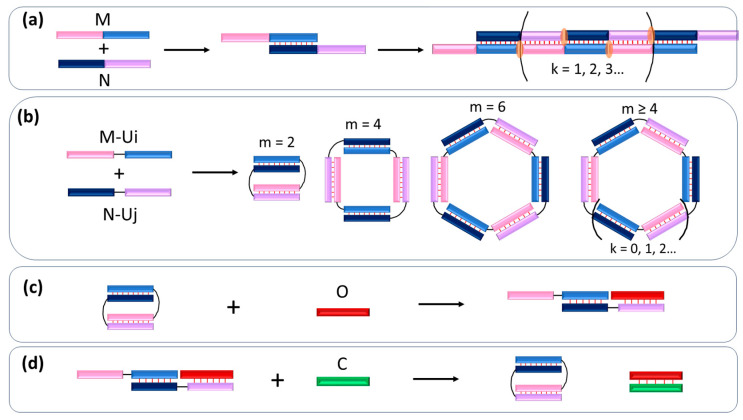
Schematic representation of the complexes’ formation. (**a**) Concatemer complex; (**b**) self-limited complexes with different duplex numbers; (**c**) formation of linear trimolecular complex from self-limited complex by adding an opener (O); (**d**) formation of self-limited complex from linear with the opener by adding a closer (C). The pink block is complementary to the violet block, the light-blue block is complementary to the dark-blue block. M is the name of an oligonucleotide containing pink and light-blue blocks, and N is the name of an oligonucleotide containing violet and dark-blue blocks. M-Ui and N-Uj are the names of sequences M and N with a linker length of i and j uridines consequently. Formation of complementary base pairs schematically marked by thin red lines. Base staking interaction in nicks of concatemer is highlighted by orange.

**Figure 2 molecules-29-00010-f002:**
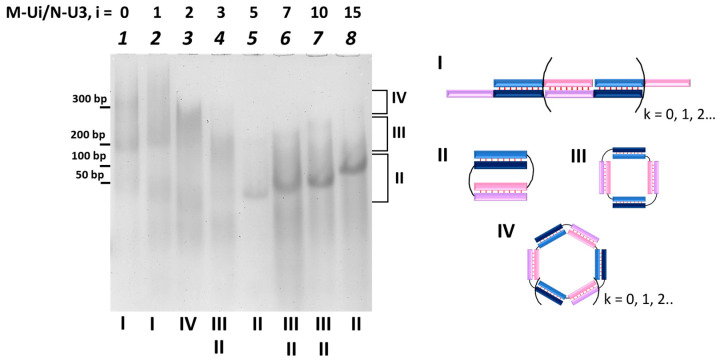
The gel shift assay of oligonucleotides’ complexes M-Ui/N-U3 with various lengths of linkers (i = 0, 1, 2, 3, 5, 7, 10, or 15). Lanes 1—M/N-U3; 2—M-U1/N-U3; 3—M-U2/N-U3; 4—M-U3/N-U3; 5—M-U5/N-U3, 6—M-U7/N-U3; 7—M-U10/N-U3; 8—M-U15/N-U3. The types of complexes are labeled below the lanes and shown on the right. A dsDNA ladder of 50–1000 bp is shown on the left. The color designations of oligonucleotides and complexes are identical to those shown in Figure 1.

**Figure 3 molecules-29-00010-f003:**
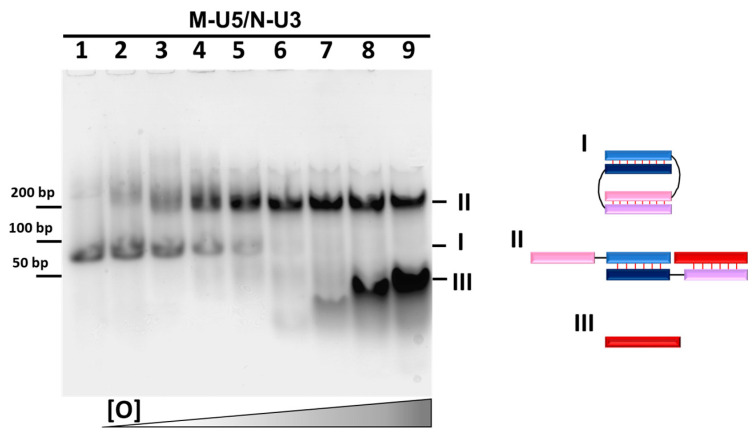
Determination of self-limited complex molecularity by adding the opener (O). Gel shift assays of M-U5/N-U3 complexes complex in the presence of RNA-opener O. Lanes: 1, M-U5/N-U3 (1:1); 2, M-U5/N-U3/O (1:1:0.1); 3, M-U5/N-U3/O (1:1:0.25); 4, M-U5/N-U3/O (1:1:0.5); 5, M-U5/N-U3/O (1:1:0.75); 6, M-U5/N-U3/O (1:1:1); 7, M-U5/N-U3/O (1:1:2); 8, M-U5/N-U3/O (1:1:5); 9, M-U5/N-U3/O (1:1:10). In the brackets the ratio of components concentration in the sample noted. Value 1 corresponded to 10 μM. The types of complexes are labeled and shown on the right. A dsDNA ladder of 50–1000 bp is shown on the left. The color designations of oligonucleotides and complexes are identical to those shown in Figure 1.

**Figure 4 molecules-29-00010-f004:**
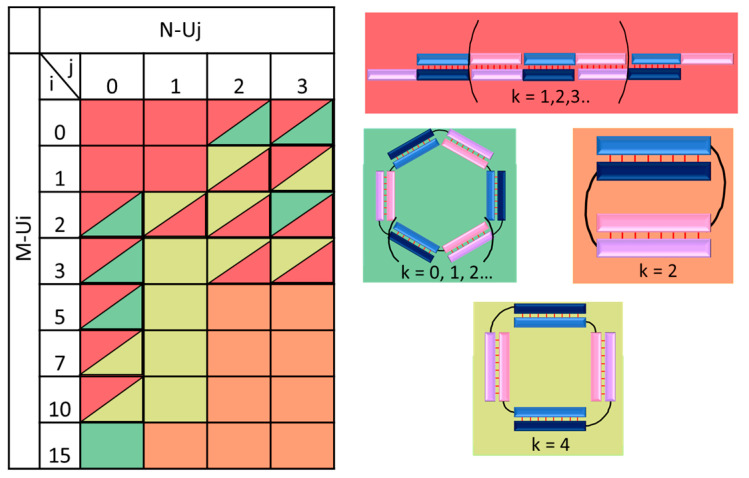
Heat map showing the types of complexes formed by a pair of oligonucleotides M-Ui and N-Uj with various linkers length: concatemer (red); bimolecular complex (salmon); tetramolecular (light green); high-molecular self-limited complex (green). If a cell has two colors, it means that the molecularity for it is not unambiguous. k is the number of duplexes in the complex; i and j is the linker length. The upper location red triangle in the cells indicates the predominant formation of concatemer, lower location—domination of self-limited complex(es). The color designations of oligonucleotides and complexes are identical to those shown in Figure 1.

**Figure 5 molecules-29-00010-f005:**
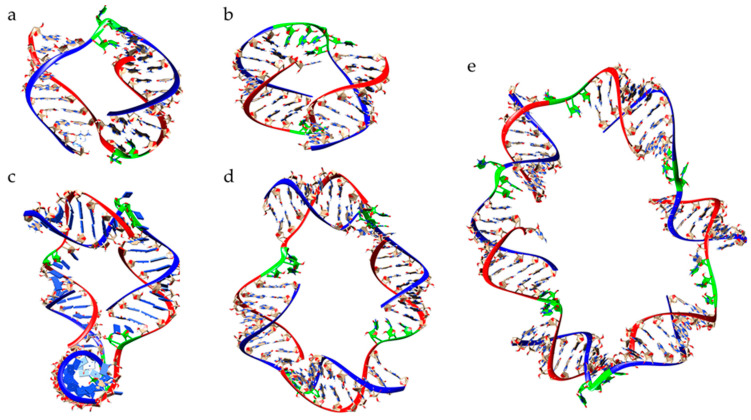
Molecular structures of the studied complexes most represented in MD trajectories: (**a**) M-U3/N-U2, (**b**) M-U5/N-U2, (**c**) (M-U3/N-U2)_2_, (**d**) (M-U3/N-U3)_2_, (**e**) (M-U3/N-U3)_6_. Linkers are shown as green, oligonucleotides of the M series are shown with blue backbone, N series with red backbone. The structures obtained by hierarchical cluster analysis.

**Figure 6 molecules-29-00010-f006:**
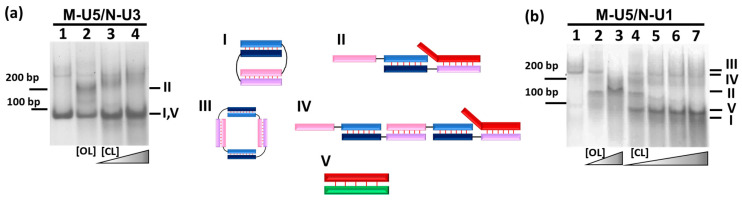
Verification of the reversible changes from cyclic complexes to linear and vice versa. (**a**) Gel shift assays of M-U5/N-U3 complex in the presence of elongated opener OL and elongated closer CL. Lanes: 1, M-U5/N-U3 (1:1); 2, M-U5/N-U3/OL (1:1:1); 3, M-U5/N-U3/OL/CL (1:1:1:1); 4, M-U5/N-U3/OL/CL (1:1:1:2). (**b**) Gel shift assays of M-U5/N-U1 complex in the presence of OL and CL. Lanes: 1, M-U5/N-U1 (1:1); 2, M-U5/N-U1/OL (1:1:1); 3, M-U5/N-U1/OL (1:1:2); 4, M-U5/N-U1/OL/CL (1:1:2:1); 5, M-U5/N-U1/OL/CL (1:1:2:2); 6, M-U5/N-U1/OL/CL (1:1:2:3); 7, M-U5/N-U1/OL/CL (1:1:2:5). In the brackets the ratio of components concentration is presented. Value 1 corresponded to 10 μM. The types of complexes are labeled on the right and illustrated in the middle. A dsDNA ladder of 50–1000 bp is shown on the left. The color designations of oligonucleotides and complexes are identical to those shown in Figure 1.

**Table 1 molecules-29-00010-t001:** Oligonucleotides code and sequence.

Code	Sequence, 5′→3′
M	CUAACUAACGCCAUCAUAUG
M-U1	CUAACUAACGUCCAUCAUAUG
M-U2	CUAACUAACGUUCCAUCAUAUG
M-U3	CUAACUAACGUUUCCAUCAUAUG
M-U5	CUAACUAACGUUUUUCCAUCAUAUG
M-U7	CUAACUAACGUUUUUUUCCAUCAUAUG
M-U10	CUAACUAACGUUUUUUUUUUCCAUCAUAUG
M-U15	CUAACUAACGUUUUUUUUUUUUUUUCCAUCAUAUG
N	CGUUAGUUAGCAUAUGAUGG
N-U1	CGUUAGUUAGUCAUAUGAUGG
N-U2	CGUUAGUUAGUUCAUAUGAUGG
N-U3	CGUUAGUUAGUUUCAUAUGAUGG
M10	CUAACUAACG
N10	CGUUAGUUAG
O	AAAAACGUUAGUUAG
C	CUAACUAACGUUUUU
OL	CCGGAAAAACGUUAGUUAG
CL	CUAACUAACGUUUUUCCGG
DO	d(CCATCATATGAAAAA)

**Table 2 molecules-29-00010-t002:** Melting temperatures (Tm, °C) ^1^ of the complexes studied.

	N	N-U1	N-U2	N-U3
M	60.7	58.6	58.3	59.6
M-U1	58.1	53.7	55.2	55.6
M-U2	57.6	53.0	54.7	53.2
M-U3	56.6	50.0	53.0	55.6
M-U5	58.7	51.1	55.4	50.0
M-U7	59.2	54.4	55.6	57.2
M-U10	58.4	51.0	56.5	51.0
M-U15	60.0	50.2	57.2	56.8

^1^ The error of Tm is ±0.5 °C.

## Data Availability

Data are contained within the article and Appendix A.

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
