# Peer review of "Diversity of Self-Assembled RNA Complexes: From Nanoarchitecture to Nanomachines"

_molecules, 2023, doi:10.3390/molecules29010010_

Round 1
Reviewer 1 Report
Comments and Suggestions for Authors
In the manuscript titled "Diversity of self-assembled RNA complexes: from nanoarchitecture to nanomachines", the authors have described the self assembled complex formation of RNA oligonucleotides employing two complementary RNA blocks connected by linkers of various lengths. They have elucidated the approach of an oligo assisted opening of the self assembled complexes to identify the molecularity of the complex. Furthermore, they elucidated the reversibility of complex formation using opener and closer oligos. The authors used Tm measurements, gel shift assay, and MD simulation to study the complex formation.
This work if a follow up of their previously reported (ref 22) study of a similar approach to study complex formation and molecularity using DNA oligonucleotides.
- Figure 1: In the captions it is mentioned that the pink is complementary to light blue. However, from the sequence provided in Table 1 and schematics pink is complementary to violet and light blue is complementary to dark-blue. Please explain.
- In results section 2.1: Have you used any theoretical calculation to design the linker lengths and/or sequence of the oligonucleotides? How is the analysis and results different between DNA (reported in ref 22) vs RNA?
- Figure 2:
Do you have a control to accurately determine the length/structure of RNA complexes? Is the length of ladder marked on left based on a DNA or RNA ladder run in the gel?
How do you confirm the formation of type II and type III complexes in lanes 4, 6, and 7?
The lanes 6 corresponds to complex M-U&/N-U3 based on figure captions and it shown to have both type II and type III complexes. However, in figure S5, the lane 15 that corresponds to the same complex is shown to have only the type II complex. Same is the case for lane 7 in figure 2 and lane 15 in figure S5. Please explain this discrepancy.
- Figure 4: Can you compare the results of the heat map to one you obtained in previous report to DNA complexes? Also, please provide any similarities, differences, and trends that you see when comparing DNA and RNA nanostructures. It would be helpful to add major finding in conclusion section as well.
- Have you used any other structural methods including AFM to look at the RNA nanostructures? It would help to accurately validate the self assembled and concatemeric complexes.
- First two tables in supporting information (used in lines 164 and 124 in main text) have the same table name "Table S1". Please update the name in SI and main text accordingly.
- Figures S7 to S13:
Wherever a dose dependent titration of opener is used, add a symbol to indicate the direction of increasing concentration of the opener. It is a bit confusing to read the caption and go back to look at the figure to understand how adding opener affects the complex formation, especially since in the same figure the titration of two openers go from high to low and low to high.
Please double check the figure captions. In multiple places a comma is used instead of decimal point. For example, in figure S7 caption for complex M-U5/N-U1/DO (1:1:0,5).
- Line 44: please add reference.
Comments on the Quality of English LanguageMinor edits needed.
Author Response
We are grateful to you for your thorough review of our manuscript and for providing us very valuable suggestions. At this point, we have completed our revisions according to your advice and enclosed the revised version of our manuscript along with our responses to your comments.
Q1) - Figure 1: In the captions it is mentioned that the pink is complementary to light blue. However, from the sequence provided in Table 1 and schematics pink is complementary to violet and light blue is complementary to dark-blue. Please explain.
Response:
The Figure 1 caption was corrected. The pink block is complementary to the violet block, the light-blue block is complementary to the dark-blue block in accordance to the Figure 1.
Q2) - In results section 2.1: Have you used any theoretical calculation to design the linker lengths and/or sequence of the oligonucleotides? How is the analysis and results different between DNA (reported in ref 22) vs RNA?
Response:
Based on our previous result, we expected the formation of complexes of different sizes and types depending on the size of the linker and duplex-forming blocks. Here we used RNA sequences similar to DNA in ref. 22 with two 10 nt long pairwise complementary blocks connected via an oligouridine linker. This choice originated from the ability to compare DNA and complexes. To date, there is no predictive model for the formation of self-limiting complexes. Thus, we analyzed the effect of the linker in the M- and N-type chains on the type and size of the complex formed. The sequences of newly designed OL and CL oligonucleodes were chosen to avoid the formation of intramolecular or self-complementary complexes using the RNA Structure Web Server and to ensure high thermal stability of the OL/CL duplexes using nearest neighbor parameters.
Corrections in the text:
Sections 2.1 and 2.4 were corrected.
- Figure 2:
Q3) Do you have a control to accurately determine the length/structure of RNA complexes? Is the length of ladder marked on left based on a DNA or RNA ladder run in the gel?
Response:
We controlled RNA complexes by analyzing electrophoretic mobility using a double-stranded DNA ladder in every gel. The use of the DNA ladder is driven by the need to compare previously obtained data for DNA complexes (in very similar conditions) with the RNA complexes analyzed here. Two types of openers (DNA and RNA) were used to confirm the formation of the self-limited complexes. Experiments with DNA opener indicate the formation of complexes without overhangs. Experiments with RNA openers allow determining the molecularity of the complexes by analyzing the number and mobility of new bands that correspond to the linearized conformations (see next answer).
Corrections in the text:
Section 2.3 is corrected.
In the legend of figures, the information on the ladder was added and in first paragraph of Section 2.3.
Q4) How do you confirm the formation of type II and type III complexes in lanes 4, 6, and 7?
Response:
We added detailed information on the way to determining of complex types in the Section 2.3. To make it clear, we have added to Fig. 2 the location of the complex types on the electropherogram.
Corrections in the text:
To determine what type of complexes the bands with a given mobility belong to and to determine their molecularity, we conducted additional studies.
Thus, M-U5/N-U3 complexes with a mobility of 80 bp are bimolecular. Complexes with similar mobility and longer linkers, M-U7/N-U3, M-U10/N-U3, and M-U15/N-U3 (Figure 2, lanes 6-7), having a mobility of 90–100 bp similarly corresponded to bimolecular complexes. The presence of a lower mobility band with a cut band at 220 bp indicates the absence of overhangs, allowing the formation of concatemers with diffuse filled lanes. This confirms the formation of self-limiting tetramolecular complexes similar to those demonstrated for M-U5 and N-U1. For the M-U2/N-U3 complex, a highly diffuse band with a mobility of ~300 bp was observed. The presence of a band with a sharp boundary at the top indicates the formation of a high-molecular complex with low stability (Fig. 2, lane 3, type IV complex).
Q5) The lanes 6 corresponds to complex M-U7/N-U3 based on figure captions and it shown to have both type II and type III complexes. However, in figure S5, the lane 15 that corresponds to the same complex is shown to have only the type II complex. Same is the case for lane 7 in figure 2 and lane 15 in figure S5. Please explain this discrepancy.
Response:
We double-checked the mobility and complex types of all the studied samples. The presence of two bands (one of them in a minor amount) corresponded to the formation of two types of complexes. The minor quantity of the tetramolecular complexes can be found in the electropherogram. Table S2 (Figure S5) was corrected.
Q6) - Figure 4: Can you compare the results of the heat map to one you obtained in previous report to DNA complexes? Also, please provide any similarities, differences, and trends that you see when comparing DNA and RNA nanostructures. It would be helpful to add major finding in conclusion section as well.
Response:
Comparison of DNA and RNA complex types based on the nucleotide linker length added in Section 2.3.
Corrections in the text:
The possible types of complexes, depending on the length of the linker, are the same for DNA and RNA. For example, concatameter complexes are formed in the absence of linkers in both chains or with a size of linker equal to 1 nt. If one of the chains has a linker that is two or more nt long, in the case of RNA, a mixture of concatameric and self-terminating complexes is often formed. In this case, DNA usually forms self-limiting complexes with a molecularity that depends on the length of the linker. Samples in which predominantly one type of complex for RNA is formed are M-Ui/N-Uj, with i five and over and j over one. Typically, bimolecular complexes of DNA and RNA are formed if the linker in the M-chain chain has a length of five or more nucleotides and two or more nucleotides in the N-chain. This is enough to ensure that there are no steric hindrances to the formation of a V-shaped structure.
Q7) - Have you used any other structural methods including AFM to look at the RNA nanostructures? It would help to accurately validate the self assembled and concatemeric complexes.
Response:
We haven’t use AFM for analysis of the analysis. Typically for RNA we observe bi- and tetramolecular complexes. It would be very difficult to identify difference between them using AFM. We did not found any other suitable methods for determining size of the complexes (DLS, mass-spectrometry, NMR, fluorescence, ligation assays, etc). The only EPR (using spin labeling in the linkers) allows one to confirm complexes types by measuring interspin distances in DNA complexes with the spin labels introduced into the linker phosphate residue. For this purpose, we used pulsed electron-electron double resonance (PELDOR, also frequently called DEER) in combination with enhanced molecular dynamics simulation. Paper was send to PCCP journal.
Q8) - First two tables in supporting information (used in lines 164 and 124 in main text) have the same table name "Table S1". Please update the name in SI and main text accordingly.
Response:
The Tables names were corrected.
- Figures S7 to S13:
Q9) Wherever a dose dependent titration of opener is used, add a symbol to indicate the direction of increasing concentration of the opener. It is a bit confusing to read the caption and go back to look at the figure to understand how adding opener affects the complex formation, especially since in the same figure the titration of two openers go from high to low and low to high.
Response:
Figures S7 to S13 corrected
Q10) Please double check the figure captions. In multiple places a comma is used instead of decimal point. For example, in figure S7 caption for complex M-U5/N-U1/DO (1:1:0,5).
Response:
Figures captions were checked and corrected.
Q11) - Line 44: please add reference.
Response:
Reference added.
Reviewer 2 Report
Comments and Suggestions for Authors
In the manuscript by Kanarskaya et al., the authors developed a new tool for RNA nanotechnology by demonstrating the spontaneous formation of various self-limited and concatemer complexes through oligoribonucleotides, and highlight its potential applications in the rational design of therapeutic nucleic acids with high target specificity and programmable immune system response, utilizing a toehold-mediated approach for complex construction and modification. Here are some comments.
Major:
1. For the gel images in Figures 2, 3, and 6, instead of simply using lane numbers plus a description for each lane in the legends, the authors should directly put the conditions for the lanes above the gel images, e.g., using condition description/reagent names, and “+”, “-”, and others, to make the figures easier to read, even without checking the legends. The same issues are also for some supplementary figures with gel images.
2. For the gel image in Figure 2, the authors should label each band as much as possible as done in Figure 3. The gel image in Figure 6a, although with some band labels, it is unclear which band belongs to II and the authors should label and explain the different bands around II.
3. How do the authors decide which band is for each type of complex, especially for those circular complexes (e.g., complexes III and IV in Figure 2), whose migration rates should be different from linear ones (e.g., complex I in Figure 2)?
4. The oligonucleotides used in this study are short (10 nt for each block). Have the authors tried longer oligos to see if they can self-assemble? If so, what is the longest limit? Likewise, have the authors tried changing the sequence of the oligos/blocks and do the same experiments? Because RNA structures are so flexible, I wonder if the results can be reproduced when longer oligos or different sequences are used.
5. Lines 165-176, based on Figure S7 that shows unclear distribution, it is inconclusive whether the DNA opener interacts with RNA complexes.
Minor:
1. Lines 61-62 should be changed to “The proposed approach for the self-assembly of linear concatemer 1D and self-limited 2D complexes can be transferred to RNA nanostructures.”
2. Lines 147-148 are unclear and need modifying.
Comments on the Quality of English Languageminor issues as shown above.
Author Response
We are grateful to you for your thorough review of our manuscript and for providing us very valuable suggestions. At this point, we have completed our revisions according to your advice and enclosed the revised version of our manuscript along with our responses to your comments.
Major:
Q1) For the gel images in Figures 2, 3, and 6, instead of simply using lane numbers plus a description for each lane in the legends, the authors should directly put the conditions for the lanes above the gel images, e.g., using condition description/reagent names, and “+”, “-”, and others, to make the figures easier to read, even without checking the legends. The same issues are also for some supplementary figures with gel images.
Response:
Figures in the main text and in the Supplementary information corrected.
Q2) For the gel image in Figure 2, the authors should label each band as much as possible as done in Figure 3. The gel image in Figure 6a, although with some band labels, it is unclear which band belongs to II and the authors should label and explain the different bands around II.
Response:
Figure 3 is corrected and explanation in the text added (see response for Reviewer 1 Question 4).
Q3) How do the authors decide which band is for each type of complex, especially for those circular complexes (e.g., complexes III and IV in Figure 2), whose migration rates should be different from linear ones (e.g., complex I in Figure 2)?
Response:
We additionally discuss this point in the text (see response for Reviewer 1 Question 6).
Corrections in the text:
To determine what type of complexes the bands with a given mobility belong to and to determine their molecularity, we conducted additional studies.
Thus, M-U5/N-U3 complexes with a mobility of 80 bp are bimolecular. Complexes with similar mobility and longer linkers, M-U7/N-U3, M-U10/N-U3, and M-U15/N-U3 (Figure 2, lanes 6-7), having a mobility of 90–100 bp similarly corresponded to bimolecular complexes. The presence of a lower mobility band with a cut band at 220 bp indicates the absence of overhangs, allowing the formation of concatemers with diffuse filled lanes. This confirms the formation of self-limiting tetramolecular complexes similar to those demonstrated for M-U5 and N-U1. For the M-U2/N-U3 complex, a highly diffuse band with a mobility of ~300 bp was observed. The presence of a band with a sharp boundary at the top indicates the formation of a high-molecular complex with low stability (Fig. 2, lane 3, type IV complex).
Q4) The oligonucleotides used in this study are short (10 nt for each block). Have the authors tried longer oligos to see if they can self-assemble? If so, what is the longest limit? Likewise, have the authors tried changing the sequence of the oligos/blocks and do the same experiments? Because RNA structures are so flexible, I wonder if the results can be reproduced when longer oligos or different sequences are used.
Response:
A real problem for analyzing the properties of long RNAs is that they form stable secondary structures. When considering longer complexes, assembly will be a problem, and a large number of off-target complexes due to variations in secondary structures is very likely. Such a problem will have both a thermodynamic and kinetic nature. Thus, the design of such structures is itself a huge and very interesting task. We will definitely try to do this in the future. Thank you very much for the recommendation.
Q5) Lines 165-176, based on Figure S7 that shows unclear distribution, it is inconclusive whether the DNA opener interacts with RNA complexes.
Response:
Figure S7 shows that there are no changes in the distributions of the band(s) corresponded to the self-limited complexes on the electropherogram after adding up to a 10-fold excess of DO. This indicates that DNA does not interact with the complex(es) in the electropherograms presented. This was further confirmed by studying the thermal stability of the complexes. The melting point of the DNA/RNA complex (N-U3/DO - 44 °C) is lower than that of RNA/RNA (M10/O - 51.8 °C), which further indicates that DNA is not able to displace one of the RNA chains as part of a self-limited complex.
The DNA opener position in Figure S7 is marked.
Minor:
Q6) Lines 61-62 should be changed to “The proposed approach for the self-assembly of linear concatemer 1D and self-limited 2D complexes can be transferred to RNA nanostructures.”
Response:
The text is corrected.
Q7) Lines 147-148 are unclear and need modifying.
Response:
The text is corrected.
Corrections in the text:
Previously, we have shown that the molecularity of the complexes determined by gel shift assay should be verified [15]. This is due to the fact that the mobility of a self-limiting complex depends not only on the size of the complex but also on a number of other factors.
Round 2
Reviewer 1 Report
Comments and Suggestions for Authors
The authors have addressed all the questions and made necessary edits to the manuscript.
Reviewer 2 Report
Comments and Suggestions for Authors
the authors have largely addressed my concerns.